# WiNeRT: Towards Neural Ray Tracing for Wireless Channel Modelling and Differentiable Simulations

**Tribhuvanesh Orekondy, Kumar Pratik, Shreya Kadambi, Hao Ye,**
**Joseph Soriaga**, **Arash Behboodi**
Qualcomm AI Research*

## Abstract

In this paper, we work towards a neural surrogate to model wireless electromagnetic propagation effects in indoor environments. Such neural surrogates provide a fast, differentiable, and continuous representation of the environment and enables end-to-end optimization for downstream tasks (e.g., network planning). Specifically, the goal of the paper is to render the wireless signal (e.g., time-of-flights, power of each path) in an environment as a function of the sensor's spatial configuration (e.g., placement of transmit and receive antennas). NeRF-based approaches have shown promising results in the visual setting (RGB image signal, with a camera sensor), where the key idea is to algorithmically evaluate the 'global' signal (e.g., using volumetric rendering) by breaking it down in a sequence of 'local' evaluations (e.g., using co-ordinate neural networks). In a similar spirit, we model the time-angle channel impulse response (the global wireless signal) as a superposition of multiple paths. The wireless characteristics (e.g., power) of each path is a result of multiple evaluations of a neural network that learns implicit ray-surface interaction properties. We evaluate our approach in multiple indoor scenarios and demonstrate that our model achieves strong performance (e.g., $<0.33$ns error in time-of-flight predictions). Furthermore, we demonstrate that our neural surrogate whitens the 'black-box' wireless simulators, and thus enables inverse rendering applications (e.g., user localization).

## 1 Introduction

Realistic simulations of physical processes are vital to many scientific and engineering disciplines. In this paper, we focus on simulation of *wireless* electromagnetic (EM) signals within a propagation environment. The physics of such EM wave propagation between a transmit and receive point are analytically given by Maxwell equations: the transmitted wave undergoes different interactions with the environment (e.g., reflection), and the receiver gets the wave through multiple *paths* with different time-of-flights and powers, and from different directions. However, solving the Maxwell equations with boundary conditions requires in-depth knowledge of the propagation environment, hence classically modelling EM propagation is intractable for most engineering applications.

Existing techniques make such simulations tractable by trading-off accuracy for speed. At one end of the spectrum, such simulations are represented in a statistical sense where a probabilistic model roughly captures the marginalized distribution over time-of-flights, gains and direction of transmit-receive paths. However, this level of accuracy is insufficient for designing systems that efficiently operate in high frequency bands. This motivates solutions at the other end of the spectrum: wireless ray tracing simulators. Given a detailed CAD representation of the environment along with the material properties, and numerous wireless configuration parameters (e.g., placement of a base station), the simulators generate resulting propagation characteristics.

Although wireless ray tracing simulators are appealing, there are a few drawbacks. First, they are generally slow, which poses a bottleneck for closed-loop design pipelines, as wireless configurations cannot be quickly mapped to propagation characteristics. Second, because they are non-

---

*Qualcomm AI Research is an initiative of Qualcomm Technologies, Inc

differentiable, they are not amenable with inverse physical design formulations, for example optimizing base station placement with the simulator in the optimization loop. Third, they usually require additional fine-tuning with real data as they are not data-driven. Calibrating them with real-world measurements is non-trivial and tedious. Fourth, they cannot generally inter-operate with probabilistic frameworks which have the advantage of better dealing with epistemic uncertainties. We believe *neural* surrogates provide a natural solution to circumvent many of these drawbacks of classical ray tracing simulators.

In this work, we propose a neural wireless simulator ('WiNeRT') by building on recent advances in scenes representation as continuous-function neural networks (Sitzmann et al., 2019; Tancik et al., 2020; Mildenhall et al., 2020). In particular, central to our approach is learning a network to model ray-surface interactions, i.e., the network transforms an incident wireless ray to an attenuated outgoing ray. By shooting out a number of rays and evaluating the network at relevant spatial regions in the environment, we estimate the wireless characteristics as a set of transmit-receive paths, each path encodes attributes such as time-of-flight and gain. Our approach also addresses some unique technical challenges posed by the non-visual wireless modality, such as dealing with sparse high-dimensional time-angle measurement signals.

We demonstrate that our neural wireless simulator reasonably renders the wireless propagation aspects by evaluating on two datasets which captures 50-100 m$^2$ indoor propagation scenes. Interestingly, we find that the 3D-structure-aware implicit formulation is a strong inductive bias and helps generalization to significant inference-time distributions shifts. Finally, we demonstrate the potential of our differentiable forward model in solving inverse problem by tackling the user localization problem after posing it as an inverse rendering problem. Our results indicate that simulator physics for specified environments can be 'distilled' into neural surrogates and thereby presenting first steps towards closed-loop design pipelines of wireless communication systems.

## 2 RELATED WORK

**Physics-based Neural Simulations.** There exists a wide body of literature to model physical processes using advances in neural networks (Djeumou et al., 2022; Karniadakis et al., 2021; Raissi et al., 2017). As simulating physical processes can be expensive and can also present non-differentiable 'black-box' in design pipelines, recent literature addresses how to work towards neural surrogates, such as for particle simulation (Sanchez-Gonzalez et al., 2020), mesh simulations (Pfaff et al., 2020), design of particle accelerators (Shirobokov et al., 2020), and inverse kinematics (Sun et al., 2021). In this paper, we are particularly interested in a specific physical process – wireless EM-wave propagation. Although this has received limited recent attention (Xia et al., 2020) in a 3D-oblivious setting, it is unclear whether these extend to complex configurations. Consequently, in this work, we work towards the first 3d-structure-aware surrogates for wireless ray tracing simulation.

**Neural Channel Modelling.** Although propagation channel modeling has been a central topic in wireless communication (Jakes & Cox, 1994; Lee, 1982; Rappaport et al., 2022), there has been a recent trend for fully data-driven models. The main paradigm of these activities is to use machine learning to learn complex distributions, model non-linearities and have differentiable simulators. These works can be categorized as statistical channel models where the channel input-output relation is modelled as a conditional probability distribution. Many works leverage recent advances in generative modelling and use models like generative adversarial networks (GANs) (Goodfellow et al., 2014) or variational autoencoders (VAEs) (Kingma & Welling, 2013) to learn the channel model (O'Shea et al., 2019; Ye et al., 2018; Yang et al., 2019; O'Shea et al., 2019; Orekondy et al., 2022; Ye et al., 2020; Dörner et al., 2020). In contrast to these works, our approach inscribes within ray tracing channel modeling paradigm, where wireless propagation is precisely modelled by tracing wireless rays, however, unlike classical ray tracers, our model is able to blend in the elements of statistical modeling and is trainable directly on field data. To the best of our knowledge, this work is the first differentiable neural ray tracer for wireless channel modelling.

**Neural Scene Representations.** Representing scenes (or more generally signals) has been widely studied in literature, such as encoding the signal in the latent space of a generative model (Kingma & Welling, 2013; Goodfellow et al., 2014). A more recent link of work encodes the signal in the parameters of a co-ordinate MLP (Park et al., 2019; Sitzmann et al., 2020; Tancik et al., 2020; Fathony

**Figure 1: Approach Overview.** We learn a forward simulator $\mathtt{render}_\theta(\cdot)$ that maps an environment configuration to a wireless channel $\boldsymbol{h}_i$. Here, $\boldsymbol{h}_i$ is a set of wireless propagation paths between $\boldsymbol{x}_{\text{tx}}$-$\boldsymbol{x}_{\text{rx}}$ (green rays in right image), each path encoding certain channel attributes e.g., path gain.

et al., 2020), thereby mapping co-ordinates (e.g., spatial, temporal) to the signal intensity values (e.g., pixel intensity, amplitude). In a specific case where the signal is a 2D RGB image, recent works (Schwarz et al., 2020; Niemeyer & Geiger, 2021; Mildenhall et al., 2020) show promising results by additionally employing image-based differentiable rendering paradigms (Drebin et al., 1988; Liu et al., 2019) to recover 3D properties of the scene. Inspired by this idea, our work neurally represents a *wireless scene* by tackling a set of orthogonal challenges, such as dealing with sparse high-dimensional signals and particularly modelling reflection and transmission effects. Consequently, we work towards the first 3D-aware neural 'wireless' scene representation model.

## 3  APPROACH

In this section, we begin with some preliminaries to the subsequent formulation of the neural wireless ray tracing problem. We then provide an initial overview of our approach in Sec. 3.1 and then dive deeper into specific technical aspects of wireless ray marching in Sec. 3.2.

**Preliminaries: Wireless Channels** Scattering, reflection and diffraction are among the main effects in electromagnetic propagation. A general mathematical description of a wireless channel, seen as linear time varying system, is given by its impulse response Tse & Viswanath (2005); Rappaport (1996). A general model can be written as (Samimi & Rappaport, 2016):

$$h(t, \boldsymbol{\Theta}, \boldsymbol{\Phi}) = \sum_k a_k(t)\delta(t - \tau_k(t))\delta(\boldsymbol{\Theta} - \boldsymbol{\Theta}_k(t))\delta(\boldsymbol{\Phi} - \boldsymbol{\Phi}_k(t)) \tag{1}$$

where $a_k(t)$ is the complex gain, $\tau_k(t)$ is the delay (time-of-flight) of path $k$, $\boldsymbol{\Theta}_k(t)$ is azimuth and elevation angle of departure (AoD), and $\boldsymbol{\Phi}_k(t)$ is azimuth and elevation angle of arrival (AoA). Going forward, we use $\boldsymbol{\phi}_k = (\boldsymbol{\Theta}_k, \boldsymbol{\Phi}_k)$ as a shorthand to collectively represent all angles. Intuitively equation 1, represents each path as a dirac function in time-angle space. The task of channel modeling can, therefore, be reduced to predicting channel attributes $(a_k(t), \tau_k(t), \boldsymbol{\phi}_k(t))$ for a given environment map, and a transmit and receive location. See Sec. A.1 for a detailed discussion.

**Forward Model: $\mathtt{render}_\theta$.** The general goal of our forward model is to run a wireless ray simulation given a certain configuration of the propagation environment. More specifically, as shown in Figure 1, the model takes three configuration parameters as input: a 3D representation of the environment $\boldsymbol{F}$ and the spatial co-ordinates of the transmitter $\boldsymbol{x}_{\text{tx}}$ and receiver $\boldsymbol{x}_{\text{rx}}$ devices. The model predicts the wireless scene as:

$$\hat{\boldsymbol{h}} = \{\boldsymbol{u}\}_{k=1}^K = \{(a_k, \tau_k, \boldsymbol{\phi}_k)\}_{k=1}^K = \mathtt{render}_\theta(\boldsymbol{x}_{\text{tx}}, \boldsymbol{x}_{\text{rx}}, \boldsymbol{F}) \tag{2}$$

where the output is a variably-sized set of $K$ paths. Each path $\boldsymbol{u}_k$ encodes three channel attributes: gain $a_k$, time-of-flight $\tau_k$ and angles $\boldsymbol{\phi}_k$. With these predicted channel attributes, we can obtain a time-angle impulse response (the 'channel') to characterize the wireless propagation effects.

**Key Idea: Implicit Representation Network $\boldsymbol{f_\theta}$.** Our approach recursively constructs the channel by using a learnt function $f_\theta : \boldsymbol{F} \times \boldsymbol{u}_k^{(r)} \mapsto \boldsymbol{u}_k^{(r+1)}$ As shown in Figure 1, given an initial ray $\boldsymbol{u}_k^{(r=0)}$, we model the final state as an evaluation of interactions that the ray undergoes with the environment $\boldsymbol{F}$. Intuitively, $f_\theta$ models the *local* interaction of any given ray $k$ either in free-space, or in particular when it is incident on an interacting surface. In the latter case of ray-surface interaction, we leverage a co-ordinate MLP to predict the transformation (e.g., attenuation, rotation) to the incident ray.

**Representing Environment $\boldsymbol{F}$.** We primarily focus on indoor propagation environments in this paper, where the environment is a 3D geometric representation. Specifically, we consider the environment represented as a 3D mesh composed of $F$ faces and $V$ vertices, where each face corresponds

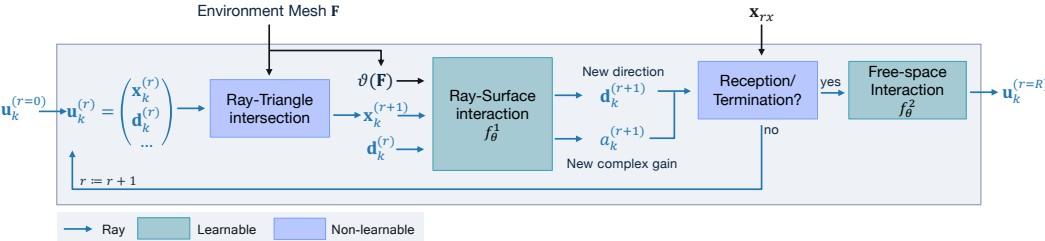

**Figure 2: Renderer: Ray Marching Steps.** At each step $r$ of the simulation, we learn the transformation introduced on a ray $\boldsymbol{u}_k^{(r)}$ e.g., reflection off a particular surface. The final transformation is a result of learnt (green blocks) and non-learnable (blue blocks) evaluations.

to some surface on a wall. We consider a mesh structure with two subtleties: (a) we represent walls as a flattened polygon and thereby do not explicitly consider its thickness; and (b) we do not encode materials of the corresponding wall faces, but rather learn the properties implicitly from data.

## 3.1 OVERVIEW: NEURAL RENDERING

In this section, we present an overview of the three steps in our approach (as shown in Fig. 1).

**Ray Launching.** We begin by shooting out a fixed set of $K$ rays from the transmitter location $\boldsymbol{x}_k^{(r=0)} := \boldsymbol{x}_{\mathrm{tx}} (\forall k)$. We launch the rays omni-directionally from the transmitter co-ordinate, agnostic to the environment and location of the receiver location. Direction $\boldsymbol{d}_k^{(r=0)}$ of each ray is oriented in the direction of a unique vertex of a ico-sphere centered at $\boldsymbol{x}_{\mathrm{tx}}$. We use the number of sub-divisions of the ico-sphere to trade-off between computational complexity and accuracy.

**Ray Marching.** The crux of our approach involves 'marching' the ray and accounting for interactions (e.g., transmission) with various surfaces of the environment. A key aspect here is using a neural network to make *local* evaluations: mapping an incident ray with some direction and power to an updated outgoing attenuated ray. The neural network is hence tasked to learn a complex non-linear characterization of the surface materials at a spatial co-ordinate. We further elaborate on the ray marching procedure in the next section.

**Ray Aggregation and Reception.** Of the $K$ rays launched from the ray launching step, we are now interested in the subset of the rays that impinges on the receiver. We model the reception sphere with a specific radius, which can be tuned to achieve a desired level of precision. To mitigate double-counting of received rays, we filter rays by associating them with a unique interaction path.

## 3.2 RAY MARCHING

We now dive deeper into the ray marching step, which tracks the evaluation of each ray as it propagates in the environment and hits various surfaces. We walk through the steps as shown sequentially in Fig. 2. We begin with a set of geometric rays $\boldsymbol{u}_k^{(r=0)}$, originating at the transmitter co-ordinate $\boldsymbol{x}_{\mathrm{tx}}$. In addition to the channel attributes of each ray (see Eq. 2), we also consider in this section an additional set of meta-attributes (e.g., origin $\boldsymbol{x}_k$, direction $\boldsymbol{d}_k$) that helps us with the ray marching steps (elaborated in Sec. A.2).

**Ray-Environment Intersections.** For each ray, we evaluate its first interaction with the environment (e.g., first wall it hits). Representing the ray geometry as $\boldsymbol{p}(t) = \boldsymbol{x}_k^{(r)} + t\boldsymbol{d}_k^{(r)}$, we are primarily interested in a solution $t > 0$ for which the ray is incident on some surface. This location helps us determine the relay (i.e., new origin) $\boldsymbol{x}_k^{(r+1)}$ for the subsequent step.

**Ray-Surface Interaction.** While the previous step solves for *where* the ray is incident in the environment, a crucial next step is determining attributes of the outgoing ray as a result of this interaction. We specifically focus on determining two attributes in this step: the new direction $\boldsymbol{d}_k^{(r+1)}$ and gain $a_k^{(r+1)}$. Popular non-neural simulators, such as Remcom (2022), look-up frequency-

dependent material properties (e.g., conductivity, permittivity) at the incidence point from standard databases (ITU-R P.2040-2) to calculate the attributes of the outgoing ray. However, it is unclear how to calculate the attributes with imprecise knowledge of the surfaces (e.g., unknown thickness and material types of each layer in a wall) or when the material properties of a layer have not been previously empirically analyzed. Our solution is to instead predict the attributes using learnt network as a function of the incident location $x_k^{(r+1)}$ and direction $d_k^{(r)}$ (see $f_\theta^1$ in Fig. 2). The ray-surface interaction network $f_\theta^1$ used in our experiments is a ReLU MLP with 3 layers (with 64-hidden units). Similar to NeRF (Mildenhall et al., 2020), we split the network into learning incident direction-independent and dependent features by concatenating direction $d_k^{(r)}$ with bottlenecked outputs of the penultimate layer in the network (See Sec. A.3 fore more details). The network predicts an attenuation factor $s$ and a rotation matrix $A$ (4-dim Euler-Rodrigues parameterization), which is then used to determine the updated gain ($a_k^{(r+1)} = s a_k^{(r)}$) and direction ($d_k^{(r+1)} = A d_k^{(r)}$).

**Reception/Termination check.** For some special cases, we halt ray marching for a subset of rays. Namely, when ray $k$ impinges on a reception sphere of a pre-specified radius (30cm in our experiments). This prevents a future version of the already received ray being potentially incorrectly received at a future iteration. In addition, for computation reasons, we also terminate ray marching if the ray exits the region of interest (e.g., ray exiting the environment).

**Free-space interaction.** While the previous steps modeled the interaction of *material* properties of the environment on wireless propagation, we now switch focus to free-space. In this case, we model propagation of a ray using the empirically-adjusted Friis' Equation: $P_r(d) = P_t G \left(\frac{d_0}{d}\right)^\lambda$ ($d \geq d_0$) which represents the power at the received at the receive antenna $P_r$ as a function of the power fed into transmitting antenna $P_t$, and the distance travelled by the ray $d$. We learn the remaining scalar parameters $G$ (antenna gain constant), $\lambda$ (attenuation factor), and $d_0$ (reference distance).

## 3.3 Training

Over the previous sections, we walked through our approach on predicting a channel $\hat{h} = \texttt{render}_\theta(x_{\text{tx}}, x_{\text{rx}}, F)$. We train the model in a supervised setting, with ground-truth time-angle impulse response measurements. Importantly, we rely only on final measurements (i.e., at $r = R$) for training and do not use any intermediate information (e.g., interaction data through a ray tracer).

**Set-based Channel Loss.** We compare two sets of multi-path channels: predictions $\hat{h} = \{\hat{u}_k\}_{k=1}^K$ and ground-truth $h = \{u_l\}_{l=1}^L$ to provide a supervisory signal for training. We evaluate the set-based loss as: $\mathcal{L}_{\text{chan}}(h, \hat{h}) = \sum_l d(u_l, \hat{u}_{\Pi(l)})$, which has two key ideas: (a) correspondence $\Pi$: we associate each ground-truth path $u_l$ with a predicted path $\hat{u}_k = \Pi(l)$. To perform such an association, we use direction-of-departure information and thereby pair paths launched in approximately the same direction; and (b) inter-path distance $d(u_l, \hat{u}_k)$: to compare two paths, we use mean square error for scalar-valued attributes (e.g., time-of-flights) and cosine distances between angular-attributes (e.g., direction of arrival). For the latter, we represent angles as unit vectors in cartesian coordinates.

**Training and Implementation Details.** We train our approach for 100 epochs using Adam optimizer with a learning rate of $10^{-3}$. We found it crucial to not aggregate rays (Sec. 3.1) in the training steps, as it led to vanishing gradients due to negligible number of rays that contributed towards gradient updates. We model the reception sphere as a fixed-sized sphere of radius 30cm. Additional implementation details are provided in Sec. C.4.

## 4 Experimental Results

In this section, we discuss experimental analysis of our neural simulator approach. We begin by discussing the preliminaries: the choice of datasets and the evaluation metrics to compare simulations. The section concludes by discussing overall performances and highlights certain benefits of neural simulations, such as running controllable simulations outside of training conditions.

## 4.1 EXPERIMENTAL SETUP: DATASETS, EVALUATION METRICS, AND BASELINES

We train and evaluate our algorithm using ground-truth data from wireless ray tracing packages. We collect two datasets, where each dataset contains channel measurements (i.e., gains, time-of-flights, angles) for different distributions of environments (e.g., floor layout). We keep the wireless configuration fixed to using omni-directional antennas at both the transmitter and receiver operating at a 3.5GHz carrier frequency. Now we further elaborate on the datasets used in our experiments.

**Dataset 1: WI3ROOMS.** We create a synthetic dataset which gives us greater control over many aspects over the generation process. Using a 10m×5m×3m hull, we randomly synthesize interior brick walls such that the eventual configuration consists of three rooms inter-connected with 1m doorways. We import the environment into an open-source wireless propagation toolbox (Amiot et al., 2013) and collect 41.6K channels, of which ∼37% of measurements are used for training.

**Dataset 2: WIINDOOR.** We use the indoor floorplans from the RPLAN dataset (Wu et al., 2019), which is popularly used to model indoor scenes (Nauata et al., 2020; 2021; Para et al., 2021). These layouts represent real-world single floor houses, with 4-8 rooms and 65-120m$^2$ areas. Each floorplan is further accompanied with room semantics such as whether a certain area is a living room, bed room, bathroom, etc. We use these semantics to selectively sample transmit/receiver locations (e.g., locations are not outside the boundary) and to determine wall materials (e.g., external facing walls are bricks, where as internal facing walls are dry plaster walls). We use a commercial ray tracer Remcom 'Wireless Insite' (Remcom, 2022) with ray tracer X3D to collect measurements in the RPLAN environment. Similar to the earlier dataset, we collect 42.5K measurements, of which ∼36% are used to train the model.

**Train and Test Regimes.** For the training dataset, we collect measurements by sampling transmitter ('Tx') from ∼10 locations (XY plane at an elevation of 2.8m) and similarly, receiver ('Rx') from 60×30 locations (but with elevation of 2m). We then create three challenging test sets (see Fig. A2 for an illustration) with novel Tx-Rx locations: (a) Checkerboard : where train and test Rx locations form a checkerboard pattern on the same XY plane at 2m elevation; (b) Generalization-$z$: where we move the test Rx locations in (a) to a novel elevation ($z$=1.0m for ThreeRooms and $z$=2.5m for RPLAN); and (c) Generalization-diag: where we sample test Rx locations on a diagonal XYZ plane. Such regimes let us validate the generalization performance under distribution shifts.

**Evaluation Metrics.** We consider three evaluation metrics to evaluate our approach: **(i) Overall prediction error ('Overall')**: We follow a similar formulation as our loss (Sec. 3.3) with one key difference – we find correspondences $\Pi$ by solving a linear-sum assignment problem. The eventual error aggregates all attributes relevant for the path (e.g., gain, angles). Intuitively, this measures the distance between two sets (sets of multi-dim paths in our case), using a similar metric common in set prediction tasks (Fan et al., 2017; Zhang et al., 2019). **(ii) Geometry prediction error ('Geometry')**: We follow a formulation similar to (i), but now focus on two specific features that captures the geometrical accuracy of the path – time-of-flight and angles at departure and arrival. Intuitively, this metric measures whether the predicted rays take the same GT route between the transmit and receive co-ordinates. **(iii) Average Delay Time - MAE ('AvgDelay')**: We average the time-of-flights $\tau_k$ per path of the channel, weighted by its linear power $p(a_k)$. We report the mean absolute error of average delays between the predicted and ground-truth channel attributes.

**Baselines.** We propose two reference baselines **(i)** $k$-**NN** (with $k$=1): which predicts the channel, given the closest match to the input spatial co-ordinates in terms of Euclidean distance **(ii) MLP**: A geometry-oblivious MLP regressor with 3-hidden layers, each with 128 units. We train the MLP using the same loss as WiNeRT. Additional details of the baselines are provided in Sec. C.4.

## 4.2 OVERALL RESULTS

In this section, we present the overall qualitative and quantitative results of our approach. We complement the overall performances with additional analysis in the next section.

**Quantitative Results.** We report the quantitative results for the two datasets (column groups) and three test sets (row groups) in Table 1. We observe from the table: (a) by focusing on the overall errors, we find WiNeRT generally outperforms all baselines, with a significant average decrease of

| | | WI3ROOMS | | | WIINDOOR | | |
|---|---|---|---|---|---|---|---|
| | | Overall | Geometry | AvgDelay | Overall | Geometry | AvgDelay |
| checkerboard | kNN | 0.232 | 0.212 | 2.238 | 0.412 | 0.396 | 2.484 |
| | MLP | 0.287 | 0.330 | 2.051 | 0.373 | 0.399 | 1.745 |
| | WiNeRT | **0.202** | **0.087** | **2.029** | **0.237** | **0.207** | **1.546** |
| gen-$z$ | kNN | 0.253 | 0.226 | 2.033 | 0.424 | 0.428 | 2.487 |
| | MLP | 0.297 | 0.350 | 1.797 | 0.388 | 0.421 | 1.969 |
| | WiNeRT | **0.217** | **0.084** | **1.522** | **0.285** | **0.250** | **1.839** |
| gen-diag | kNN | 0.252 | 0.213 | 2.118 | 0.380 | 0.251 | 1.377 |
| | MLP | 0.312 | 0.322 | 1.889 | 0.390 | 0.315 | 1.513 |
| | WiNeRT | **0.229** | **0.085** | **1.792** | **0.369** | **0.170** | **0.828** |

**Table 1: Quantitative Results.** Comparing errors of our approach (WiNeRT) with baselines, over two datasets (column groups) and three test regimes (row groups). Lower values are better and the lowest errors are in **bold**.

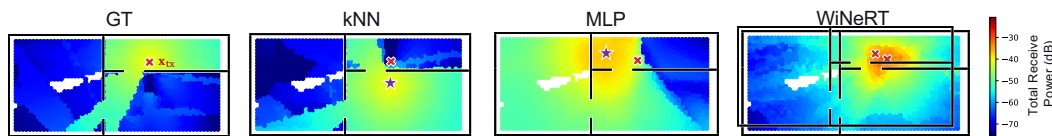

**Figure 3: Receive Powers.** By fixing the transmit location ($x_{tx}$, red cross), we measure the receive power (color at each point; in dB) predicted at each location in WI3ROOMS dataset. kNN and MLP suffer from memorization and falsely predict highest receive powers around phantom transmit locations (purple star).

-0.071 points compared to kNN and -0.085 with MLP; (b) WiNeRT is especially strong in capturing the geometry (e.g., 59-63% drop in errors w.r.t second best on WI3ROOMS) of the environment, which can be likely attributed to a strong inductive bias enforced by decoupling global rendering from local evaluations; (c) Although WiNeRT has reasonable performance in capturing the average delays, the performance gap here (e.g., 1-15% reduction in errors on WI3ROOMS) is not especially large compared to other metrics. We attribute this to contributions from 'false positive' rays with non-negligible power arising from our dense ray-launching technique. (d) The contributions of false positives can be mitigated by using a more sophisticated ray launching technique. For instance, by piggybacking on ray launch directions from GT channels, we can significantly improve performances across all metrics e.g., from 1-15% error reduction to 15-20% reduction in average delays on WI3ROOMS; (e) Overall, we attribute the underperformance of the baselines to poor generalization performance. For instance, in Figure 3, we illustrate the receive powers (in dB) predicted by all approaches in WI3ROOMS, for some placement of the transmitter (red cross in top-right room). We observe in this particular case that the high-power areas in the kNN and MLP baselines are predicted for a false phantom location (purple star), which roughly corresponds to a transmitter location in training set. This contrasts predictions by WiNeRT where the high-power areas are correctly concentrated around the transmitter location. As a result, we find that simple baselines find it challenging to generalize to new unseen spatial co-ordinates at inference time.

**Qualitative Results.** We complement the previous quanitative discussions with observations drawn from qualitative analysis. WiNeRT particularly helps for this analysis, as we can recovert intermediate ray-environment interaction information. From qualitative examples shown in Fig. 4(a, b), we draw some observations: (a) WiNeRT surprisingly learns ray-surface interactions implicitly, without any direct supervision. For instance, we observe multiple reflected paths between Tx and Rx; (b) we also find that our predictions (red rays) are generally consistent with the underlying simulation process (green rays) e.g., reflections from adjacent walls, floor and ceiling; and (c) we notice WiNeRT sometimes predicts false positives (e.g., above $x_{tx}$ in Fig. 4b), which we attribute to dense omni-directional ray launching.

### 4.3 ANALYSIS

In the previous section, we evaluated the overall performance of WiNeRT and found promising results. Now, we take a closer look at our approach and investigate generalization benefits.

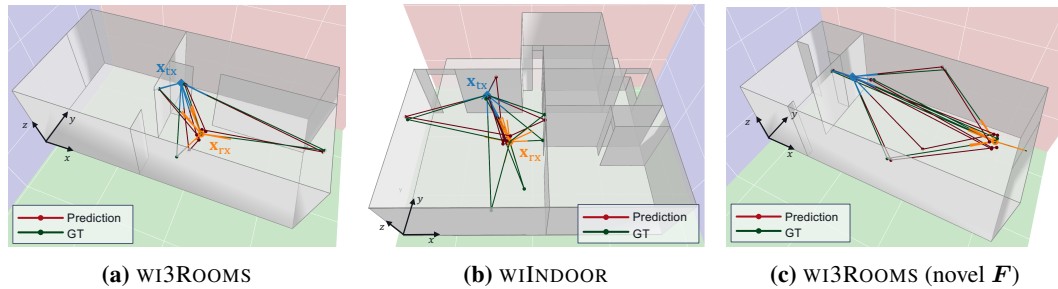

**(a)** WI3ROOMS         **(b)** WIINDOOR         **(c)** WI3ROOMS (novel $\boldsymbol{F}$)

**Figure 4: Qualitative results.** (a, b) Evaluation on WiNeRT on the environment seen during training. (c) We use the previously trained model and re-render on a re-configured floormap $\boldsymbol{F}$.

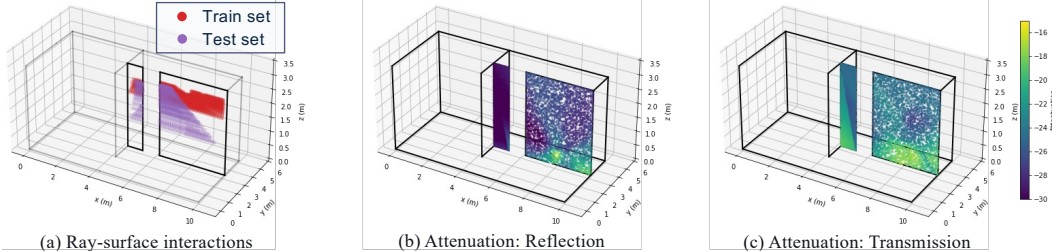

(a) Ray-surface interactions     (b) Attenuation: Reflection     (c) Attenuation: Transmission

**Figure 5: Evaluating Ray-surface interaction MLP.** We display a cut-out of the 3ROOMS represented as a wireframe, with a specific focus on a particular wall. (a) We find a train-test distribution shift of ray-surface incidence points (b, c) Evaluation of the MLP at various incidence points.

**What does the ray-surface interaction learn?** We begin by investigating the ray-surface network ($f_\theta^1$ in Fig. 2) in isolation. The network is tasked to map an incident ray (gain $a_{\text{in}}$, direction $\boldsymbol{d}_{\text{in}}$) to an outgoing ray ($a_{\text{out}}$, $\boldsymbol{d}_{\text{out}}$). To accurately make this prediction, the network needs to learn direction- and material-dependent properties at the incident location $\boldsymbol{x}_{\text{inc}}$, which poses two challenges. First, the network does not have explicit supervision to learn these properties. Rather, the network needs to *implicitly* learn these properties by optimizing over a number of channel measurements. Second, specific to our case, the measurements collected involve sparse ray-surface interactions i.e., in practise we cannot expect for paths in the training measurements to interact densely with all possible surfaces. For instance, consider Fig. 5a, which show the incident points $\boldsymbol{x}_{\text{inc}}$ for a particular wall (black edges) that we recover from the underlying ray tracing tool. Here, we observe that the implicit training set interactions (red markers; never used during our training) are localized to a ∼50cm band (15% area of the wall). However, at test-time, the network is tasked to generalize to interactions for a different distribution of incidence points (purple markers). In spite of the challenges we find the ray-surface network associates meaningful information to surface co-ordinates. For instance, we show the attenuation factor predicted for the reflected (Fig. 5b) and transmitted co-ordinates (Fig. 5c) for rays arriving from a fixed $\boldsymbol{x}_{\text{tx}}$ co-ordinate (placed at $x$=8m). We find that the network learns a smooth material- and direction-dependent function over the surface. Over the next experiments, we exploit these locally learnt properties and evaluate WiNeRT rendering in novel scenarios.

**Controllable synthesis: Predicting in Novel Environment Configurations.** The previous experiments focused on evaluating approaches for novel locations of transmit and receive co-ordinates at simulation time. Now, we consider novel test-time environments by simulating approaches on *re-configured layouts* $\boldsymbol{F}'$ of the train-time environment $\boldsymbol{F}$, such as by randomly editing placement of interior walls. Overall, we find that WiNeRT remarkably extrapolates to the reconfigured environment, with the overall error unchanged with WiNeRT (0.202 on $\boldsymbol{F}$ vs. 0.203 on $\boldsymbol{F}'$; more results in Table A2). Furthermore, by observing the results qualitatively in Figure 4c, we find the predicted interactions remain consistent with the ground-truth simulated rays in novel environment configurations. This is particularly appealing as for simulation use-cases which require modelling dynamic objects (e.g., moving vehicle), as WiNeRT allows re-configuring environment without retraining.

**Controllable synthesis: Simulating Higher-order Interactions.** In this experiment, we evaluate the ability of approaches to generalize to different numbers of interactions (denoted by $r$ in Sec. 3) at inference time. With WiNeRT, we have the ability to control the number of interactions at

test-time (i.e., by unrolling $f_\theta$ for fewer or more steps). We briefly summarize our observations here (see Table A4 for more details). WiNeRT exhibits promising results: while the baselines struggle with a simpler task of lower-order interactions (e.g., 0.22-0.58 overall errors at $r{=}0$), WiNeRT's performance improves (from 0.20 to 0.12). A better performance is natural in this particular setting, since the model is required to perform an *easier* task than original (predicting only line-of-sight component). For higher-order interactions, we observe performances of all approaches degrades, but WiNeRT outperforms the baselines. In particular, even at $r = 3$, we find the geometric-errors of WiNeRT (0.27) comparable to baselines in their originally trained setting ($r{=}1$, 0.21-0.33 errors).

**How fast are the simulations?** We investigate the wall-clock simulation times of WiNeRT and baselines and compare them with wireless ray tracers. In the specific case of WiNeRT, we have some control over the time-accuracy trade-offs at test-time by varying the density of initial rays launched (see Sec. 3.1). Overall, we find that WiNeRT demonstrates speed-ups of 11-22× over PyLayers and 6-22× over Wireless Insite. While the baselines are even faster (538-687× with MLP and 79-97× with kNN), it is achieved at the price of higher errors and poor generalization capabilities (Sec. 4.2). Overall, we find WiNeRT presents reasonable time-accuracy trade-offs compared to baselines. See Sec. C.2 for additional details.

**Exploiting differentiability: User Localization via inverse (differentiable) rendering.** Over the previous sections we focused on *forward* simulations. Now, we study a proof-of-concept for leveraging our differentiable simulator for *inverse* problems, such as for user localization: determining user location $x_{\text{rx}}$ from an observed channel $h_{\text{obs}}$. We solve for $x_{\text{rx}}$, by performing gradient on spatial coordinate $x_{\text{rx}}^{\text{ukn}}$ that minimizes the channel loss $\text{render}_\theta(x_{\text{tx}}, x_{\text{rx}}^{\text{ukn}}, F_i)$. This is possible with WiNeRT, since we can backpropagate through the neural simulation of the channel. We evaluate over 100 test examples and find encouraging results, with a median error of 0.58m in WI3ROOMS (a 150m$^3$ volume) and 1.21m in WIINDOOR (a 300m$^3$ volume). See Sec. C.4 for more details.

## 5 CONCLUSION, LIMITATIONS, AND BROADER IMPACT

In this paper, we proposed the first *neural* forward model for wireless ray tracing-based simulations. Such models are particularly appealing as they help alleviate some drawbacks of classical non-neural simulators (e.g., better handling model-measurement mismatches, non-differentiability). Towards this goal, we proposed WiNeRT which tasks an MLP to learn how surfaces in a 3D environment influence propagation of wireless rays, such as by predicting attenuation factor of a reflective component. Overall, we find promising results indicating neural simulators closely capture propagation effects. As neural simulators are additionally differentiable, we further show that they can be used to optimize inverse problems such as user localization.

**Limitations and Future Work.** This paper presents the first step towards realizing a neural surrogate for simulating propagation of wireless rays. While we find promising results – in terms of empirically mimicking the simulator's performance while simultaneously reducing complexity – many important steps remain to realize our over-arching goal of differentiable wireless ray tracing. Our approach is designed to capture linear effects of the channel in line with standards (3GPP TR 38.901; ITU-R P.2040-2) and extending to non-linear effects (e.g., amplifier saturations) remains an open-problem. Additionally, while our focus is primarily reflection and transmission properties of ray-surface interactions (capturing majority of receive power) which are increasingly relevant for high-frequency transmissions, other properties (e.g., scattering, diffraction) require investigation to model simulations across a wider radio-frequency spectrum. Finally, our surrogate's performance is currently upper-bounded by the underlying simulator's performance, motivating studies into learnt calibration of the surrogate model with real-world measurement data to bypass simulation accuracy.

**Broader Technical Impact.** Although our paper focuses on neural simulation of EM waves in the radio-frequency spectrum (0.5-100 GHz), we believe working towards this goal complements research in non-radio modalities as well. For instance, to model propagation of acoustic signals in spatial environments, estimating material-dependent ray-surface interactive properties remains a challenging problem and the proposed research direction potentially complements existing techniques. More generally, we believe that as radio signals require modelling both ray (e.g., reflection) and physical optic (e.g., interference, diffraction) properties, advances here are intertwined with many modalities across the EM spectrum (e.g., audio, visual).

REPRODUCIBILITY STATEMENT

To ensure reproducibility, we take a number of steps. On the dataset side, we use either publicly available indoor layouts (e.g., RPLAN) or synthetically generate layouts with known random seeds (0 and 10 in our case). We further elaborate on the simulation settings to recreate our dataset in Section 4.1 and Section B. We plan to release the simulated data measurements. On the implementation side, we provide specific training details in Section C.4 and further elaborate on the detailed architecture in Section A.3.

ETHICS STATEMENT

The data used in our paper corresponds to simulated data of physical processes (EM wave propagation). Since this does not involve any human subjects or personally identifiable information, we believe there is no conflict in this regard.

ACKNOWLEDGEMENT

We thank Hanno Ackermann for discussions and feedback on the paper. We additionally thank numerous colleagues for insightful discussions: Thomas Hehn, Fabio Valerio Massoli, Maziar Raissi, Afshin Abdi, June Namgoong, Taesang Yoo, and Akash Doshi.

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

# Appendix

## A    APPROACH

### A.1    BUILDING CHANNEL MODELS

This section accompanies the text in Section 3.

Channel models are defined either in a statistical way by defining a distribution over channel attributes or in deterministic way using ray tracing. Statistical channel models are inadequate for applications involving positioning, sensing and challenges of communication at higher frequencies (e.g., mmWave at 30-300 GHz (Rappaport et al., 2022)). Inspired by similar techniques in computer graphics (Glassner, 1989), traditional ray tracing approaches (see for example (McKown & Hamilton, 1991; Ikegami et al., 1991; Walfisch & Bertoni, 1988)) approximate propagation of electromagnetic waves by modeling interactions of each *ray* with objects in its paths. These interactions include for example reflection, diffraction and penetration. Although this is more efficient than solving Maxwell equations, ray tracing methods need a detailed knowledge of the environment and are generally slow for prototyping. They generally utilize hard coded and mathematically tractable models for example knife-edge model for diffraction (Lee, 1982; Rappaport, 1996). These abstractions suffer from mismatches and require occasional tedious fine-tuning and calibration with real data. Improving these models while remaining tractable for rapid simulation rounds is not straightforward. Finally, they are non-differentiable and cannot be integrated into a closed loop design pipeline. We plan to tackle these issues by building a neural surrogate of a physics-based wireless ray tracer in this paper.

### A.2    REPRESENTING RAY ATTRIBUTES

We represent the $k$-th ray (among $K$ rays) at the $r$-th iteration of rendering as $\boldsymbol{u}_k^{(r)}$. For notation convenience, we drop the sub- and super-script for the rest of the section. We characterize the wireless ray analogous to the concept of an optical ray (such as with geometric direction, intensity). In additional to the wireless attributes (see Equation 2), we further include meta-level attributes that helps us propagate and render the eventual ray received at the receiver co-ordinate $\boldsymbol{x}_{\mathrm{rx}}$. We briefly describe these attributes here and elaborate on how they are obtained or updated over the next sections. The ray contains the attributes:

$$\boldsymbol{u} = ( \underbrace{a \quad \tau \quad \phi}_{\text{(a) Channel Attributes}} \quad \underbrace{\boldsymbol{x} \quad \boldsymbol{d} \quad t_s \quad t_{rx} \quad \rho_{\mathrm{rx}}}_{\text{(b) Ray Geometry}} \quad \underbrace{\sigma_{\mathrm{upd}} \quad \sigma_{\mathrm{rx}}}_{\text{(c) State}} )$$

which as shown can be grouped into three categories: **(a) Wireless Channel Attributes.** Exactly as discussed earlier in the section (see Equation 2), it contains the attributes to construct the wireless channel time-angle impulse response (Equation 1) **(b) Ray Geometry.** We additionally include geometrical representation of the ray, which helps us determine how to propagate the ray through the environment. Specifically, we represent the geometry of the ray using the line equation: $\boldsymbol{p}(t) = \boldsymbol{x} + t\boldsymbol{d}$, where $\boldsymbol{x}$ is the origin and $\boldsymbol{d}$ is a unit-vector encoding the ray direction. We are interested in two particular solutions of $t$ in this equation: $t_s$ for which the ray intersects with a surface (mesh face in our case) and $t_{\mathrm{rx}}$ for which the ray is tangential to a sphere around some receiver of radius $\rho_{\mathrm{rx}}$. **(c) Ray state.** To help with subsequent updates to the ray at future iterations, we track two binary variables. $\sigma_{\mathrm{upd}}$ denotes whether the ray has to be updated in the next iteration. $\sigma_{\mathrm{rx}}$ denotes whether the ray has impinged on a reception sphere of a predefined radius.

### A.3    RAY MARCHING: DETAILS

**Ray-Environment Intersections.**    For each ray, we are interested in their first interaction with the environment (e.g., first wall it hits, impinging on the receiver). For this, we are interested in the solutions to the line equation representing the geometry of the ray: $\boldsymbol{p}(t) = \boldsymbol{x}_k^{(r)} + t\boldsymbol{d}_k^{(r)}$. In particular, we are interested in two solutions of $t$: **(a) Ray-Face intersection.** The smallest value of $t > 0$ for which $\boldsymbol{p}(t)$ lies on a surface (a triangular mesh face in our case). For this, we perform ray-triangle intersections with each face in the environment and find the corresponding solution $t = t_s$.

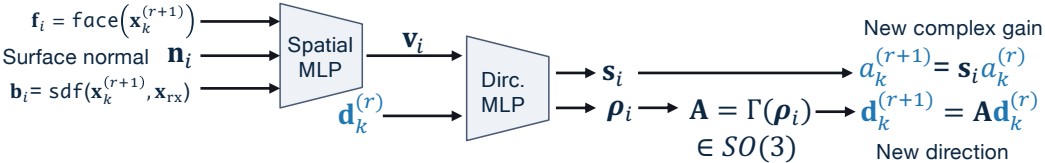

**Figure A1:** Ray-surface interaction network $f_\theta^1$

This helps us estimate the new relay location: $\boldsymbol{x}_k^{(r+1)} = \boldsymbol{x}_k^{(r)} + t_s \boldsymbol{d}_k^{(r)}$ **(a) Ray-Rx intersection.** In parallel, we are also interested in positive solutions of $t$ for which the ray hits the receiver if it were modeled as a sphere of radius $\rho_{\text{rx}}$. In this case, we obtain the value of $t$ as the projection of $\boldsymbol{x}_{rx}$ on $\boldsymbol{p}(t)$:

$$t_{rx} = \max(0, \ (\boldsymbol{x}_{rx} - \boldsymbol{x}_k^{(r)}) \cdot \boldsymbol{d}_k^{(r)}) \tag{3}$$

$$\rho_{rx} = ||(\boldsymbol{x}_{rx} - \boldsymbol{x}_k^{(r)}) - t_{rx}\boldsymbol{d}_k^{(r)}|| \tag{4}$$

Consequently, at the end of ray-environment, we analytically estimate the first intersections of the ray with both the environment and (potentially) the receiver.

**Ray-Surface Interaction.** If the ray $\boldsymbol{u}_k^{(r)}$ (originating at $\boldsymbol{x}_k^{(r)}$ and travelling in direction $\boldsymbol{d}_k^{(r)}$) hits a wall at $\boldsymbol{x}_k^{(r+1)}$ (as estimated in the previous step), we are now interested in characterizing the outgoing ray with origin at $\boldsymbol{x}_k^{(r+1)}$. Specifically, we are interested in estimating the new direction $\boldsymbol{d}_k^{(r+1)}$ (does the ray penetrate the wall? or reflect?) and the corresponding change in gain that arises (i.e., loss of power, change of phase). This is a complex problem and typically requires in-depth knowledge of the surface (e.g., which material) as well as it's specific EM properties (e.g., frequency-dependent effects). Our solution is to instead learn these properties by associating spatial regions in the environment with EM-specific properties. Towards this, we delegate the association to a neural network show in Figure A1. The key idea is to associate spatial co-ordinates (or sets of co-ordinates, given by face on which they lie) with EM properties. We achieve this by mapping spatial properties (e.g., face corresponding to $\boldsymbol{x}_k^{(r+1)}$) to EM properties (e.g., gain factor).

Specifically, our neural network is:

$$\boldsymbol{v}_i = \texttt{spatial\_net}(\boldsymbol{f}_i, \boldsymbol{n}_i, \boldsymbol{b}_i) \tag{5}$$

$$\boldsymbol{s}_i, \boldsymbol{\rho}_i = \texttt{directional\_net}(\boldsymbol{v}_i, \boldsymbol{d}_i) \tag{6}$$

which consists of a `spatial_net` to encode EM properties specific to a spatial region, but *independent* of the incidence direction. This network takes as inputs the one-hot encoding of the face $\boldsymbol{f}_i$ on which the relay point $\boldsymbol{x}_k^{(r+1)}$ lies and the surface normal vector at that point $\boldsymbol{n}_i$. In addition, we also provide the network a 3-dim conditioning vector of signed distances

$$\boldsymbol{b}_i = (\text{sdf}(\boldsymbol{x}_{tx}, \boldsymbol{f}_i), \quad \text{sdf}(\boldsymbol{x}_{rx}, \boldsymbol{f}_i), \quad \text{sdf}(\boldsymbol{x}_k^{(r+1)}, \boldsymbol{f}_i)) \tag{7}$$

where $\text{sdf}(\boldsymbol{x}, \boldsymbol{f})$ is the signed distance function between co-ordinate $\boldsymbol{x}$ and face $\boldsymbol{f}$. We find it crucial to condition the network on these values to help predict EM-properties for relevant outgoing components (e.g., reflective, transmission).

The output of the network is a gain factor $\boldsymbol{s}_i$, such that the new gain of the ray $\boldsymbol{u}_k^{(r+1)}$ is $a_k^{(r+1)} = s_i a_k^{(r)}$. Since the gain magnitudes can be represented in either linear or logarithmic scale, we predict both additive and multiplicative factors of the gain in practise ($a_k^{(r+1)} = s_{i,1} a_k^{(r)} + s_{i,2}$). In parallel, the network also predicts the rotation a ray incident with direction $\boldsymbol{d}_k^{(r)}$ on $\boldsymbol{f}_i$ undergoes. We characterize rotations using a 4-dim rotation $\boldsymbol{\rho}_i$ using Euler-Rodrigues parameterization. This parameterization encodes the axis of rotation and about which $\boldsymbol{d}_k^{(r)}$ rotates by angle $\vartheta$. We represent the rotation by a $3 \times 3$ SO(3) matrix $\boldsymbol{A}$ and the new outgoing direction of ray $k$ is given by $\boldsymbol{d}_k^{(r+1)} = \boldsymbol{A}\boldsymbol{d}_k^{(r)}$.

**Reception/Termination check.** For some special cases, we halt ray marching for a subset of rays. Namely, when ray $k$ impinges on a reception sphere of radius under $\varrho$ meters. This prevents

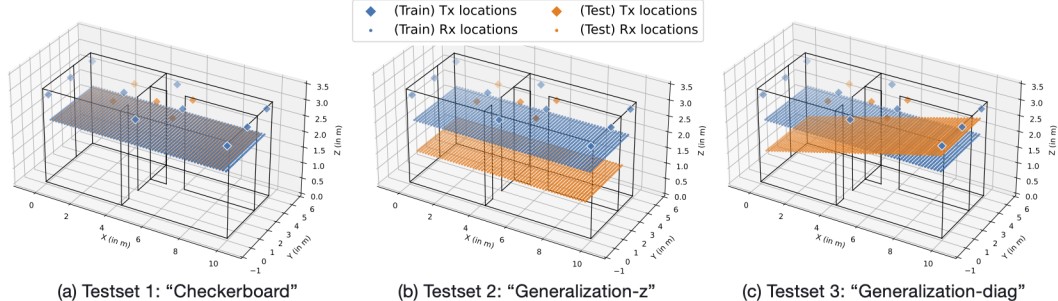

(a) Testset 1: "Checkerboard"  (b) Testset 2: "Generalization-z"  (c) Testset 3: "Generalization-diag"

**Figure A2: Train and test regimes**: We consider disjoint subsets of train (blue markers; identical in all figures) and test (orange markers) co-ordinates of transmit and receive locations.

a future version of the ray being potentially being incorrectly received once again. In addition, for computation reasons, we also terminate ray marching if the ray exits the region of interest (e.g., ray exiting the environment).

**Free-space interaction.** While the previous steps modeled the interaction of *material* properties of the environment on wireless propagation, we now switch focus to free-space. In this case, we model propagation of a ray using the empirically-adjusted Friis' Equation:

$$P_r(d) = P_t K \left(\frac{d_0}{d}\right)^\lambda, \qquad d \geq d_0 \tag{8}$$

which represents the power at the received at the receive antenna $P_r$ as a function of the power fed into transmitting antenna $P_t$ and the distance travelled by the ray $d$. We learn the remaining scalar parameters $K$ (constant representing of antenna gains), $\lambda$ (wavelength of signal), and $d_0$ (reference distance).

## B  DATASET: ADDITIONAL DETAILS

### B.1  TRAIN AND TEST REGIMES

Figure A2 accompanies the text in Section 4.1.

### B.2  SIMULATION FOR WIINDOOR DATASET: DETAILS

We created 3 different floor-plans in Wireless Insite where 2D floor-plans layout and semantic labels of each room are picked from House GAN++ dataset and mapped into a 3D layout where the scale and dimensions are determined based on practical floor-plan scenarios. All layouts are scaled to 10m×10m with ceiling height at 3m. All the inner walls and floor materials are layered dielectrics with specific permittivity, conductivity & roughness. These have finite reflection and transmission coefficients. The reflection coefficient is corrected if the surface is not smooth while the transmission coefficients are unaffected by surface roughness.

**Materials.** Propagation characteristics are naturally affected by the medium and we create a dataset with fairly diverse set of materials. Layered dielectric with two layers separated by free-space of 89cm is chosen for all inner walls and the outer-walls were made of thicker materials of concrete. Doors were created using free space except the balcony door which was created using glass with a small thickness. The balcony walls were laid out using brick walls. The propagation factor and index of reflection are functions of the permittivity ($\epsilon$) and conductivity ($\sigma$) of medium. In Table Table A1, we present the relative permittivity and conductivity.

**Antenna and Transceiver configuration.** Omnidirectional beam patterned antenna with polarization perpendicular to the z axis is setup for all receive and transmit antennas. Location, Orientation of the antenna are set relative to global reference such that they are rotated about the z axis by 90deg and placed at a height of 2.8m. All antennas employ the same configuration with no transmission loss.

|  | thickness(cm) | permittivity $\epsilon$ | conductivity $\sigma$ (S/m) |
|---|---|---|---|
| Layered drywall(1,3) | 1.3 | 2.8 | 0.013 |
| Brick | 12.5 | 4.44 | 0.0001 |
| Concrete | 30 | 5.31 | 0.015 |
| Glass | 3 | 2.4 | 0 |

**Table A1:** Material properties

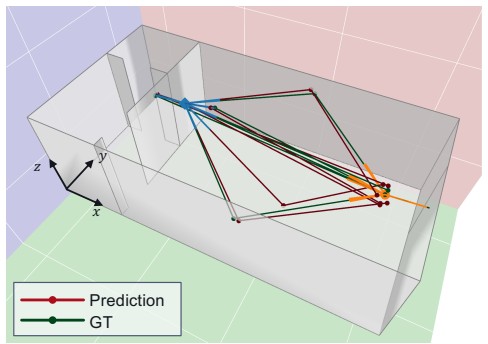

|  | Overall | Geometry | Avg. Delay |
|---|---|---|---|
| kNN | 0.264 | 0.288 | 1.479 |
| MLP | 0.280 | 0.378 | **1.191** |
| WiNeRT | **0.203** | **0.114** | 1.297 |

**Table A2: Quantitative results.** For a trained approach evaluated on a reconfigured floormap $F'$

**Table A3:** Qualitative results

**Simulation.** We currently run the simulation using the shoot and bounce model where a geometric path is drawn from every point on the transmitter field pattern to the receive point. This also includes transmission through surfaces allowing it to model transmittance and reflection. Rays are first traced from the source points with the rays reflecting specularly from the building walls. The rays that hit building walls are reflected specularly and continue to be traced up to the maximum number of reflections and transmissions.

The spatial separation of rays is set to $0.75°$. The geometric path traced by the ray undergoes upto 6 specular reflection and 3 transmittance with path loss threshold set to -70dBm.

Total received power of all paths is determined as the sum of time averaged power of group of correlated paths. A set of ray paths that interact with similar set of faces and follow nearly same path are defined as group.

## C EVALUATION: ADDITIONAL DETAILS

### C.1 CONTROLLABLE SYNTHESIS: GENERALIZATION TO RECONFIGURED FLOORMAPS

Table A2 accompanies the discussions in Section 4.3, where we evaluate a WiNeRT model trained in one environment $F$ and evaluated in a reconfigured environment $F'$.

### C.2 CONTROLLABLE SYNTHESIS: LOWER- AND HIGHER-ORDER INTERACTIONS

See Table A4, which accompanies the discussions in Section 4.3.

|  | Overall (DoD) | | | | Geometry | | | | Avg. Delay | | | |
|---|---|---|---|---|---|---|---|---|---|---|---|---|
| #interactions $r$ | 0 | 1* | 2 | 3 | 0 | 1* | 2 | 3 | 0 | 1* | 2 | 3 |
| kNN | 0.22 | 0.33 | 0.50 | 0.55 | 0.31 | 0.21 | 0.29 | 0.33 | 1.30 | 2.24 | 2.96 | 3.40 |
| MLP | 0.58 | 0.46 | 0.61 | 0.67 | 0.34 | 0.33 | 0.37 | 0.41 | 0.98 | 2.05 | 2.93 | 3.48 |
| WiNeRT | **0.12** | **0.25** | **0.44** | **0.51** | **0.00** | **0.09** | **0.21** | **0.27** | **0.03** | **2.03** | **2.43** | **2.8** |

**Table A4: Low- and Higher-Order Interactions**. We vary the number of ray-surface interactions (denoted by $r$) for a model trained using single-order interactions ($r$=1, denoted by * in the table).

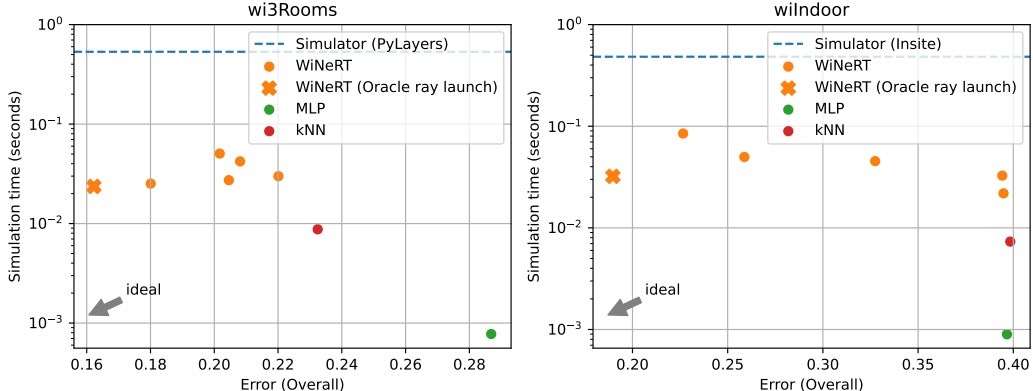

**Figure A3: Simulation Time**. Comparing wall-clock time vs. accuracy performances of our approach (WiN-eRT) against baselines (MLP, kNN) and wireless ray tracing softwares (PyLayers and Insite). The 'Oracle ray launch' variant, which utilizes known ray launch directions at test-time, indicates an approximate performance upper-bound of our approach.

## C.3 SIMULATION TIME

In Sections 4.2 and 4.3, we found our proposed approach WiNeRT achieves reasonable performance compared with non-differentiable and non-neural simulator packages. Additionally, we demonstrated that WiNeRT is capable of generalization (e.g., to novel elevations, to re-configured floorplans) and can be used for inverse problems. In this section, we additionally discuss run-time performance of WiNeRT and compare against baseline approaches as well as the simulator package.

**Experimental Setup.** The end-goal of the experiment is to analyze the simulation time (specifically wall-clock times) of the proposed WiNeRT approach and contrast it against both the simulator softwares (PyLayers, Wireless Insite) and proposed baselines (MLP, kNN). We first remark that the implementations fundamentally vary between the approaches and hence an ideal wall-clock timing comparison is not possible. For instance, some approaches (WiNeRT, MLP, kNN) use a *PyTorch* implementation which can be run on GPU whereas the wireless ray tracing simulation packages are either proprietary (e.g., Wireless Insite) or developed exclusively for CPU (e.g., PyLayers) and thereby limiting the choice of hardware on which they can be run. Nonetheless, we keep simulation settings consistent when possible: by running the exact simulations used for the overall results (setting 'checkerboard'; see Section 4.1) and furthermore estimating wall-clock times per simulation (batch size of 1) over $N$ individual simulations with a maximum of 1 reflection and transmission (i.e., $r=1$). For all approaches, we report only the mean simulation time over the multiple simulations, as we found the variances low ($\sigma^2 \leq 3.5 \times 10^{-3}$). When possible, we also report corresponding accuracy ('overall prediction error'; see Sec. 4.1). We evaluate PyTorch-based implementations (WiNeRT, MLP, kNN) over $N = \sim 8K$ simulations using pretrained models (specifically the ones for reporting 1) on a Nvidia A100 GPU. In the case of WiNeRT, we are able to control time-accuracy trade-off to some degree at test-time by varying the number of launched rays $K$ (see 'Ray Launching' in Sec. 3.1) as a function of the number of subdivisions of the ico-sphere. We choose 1-5 sub-divisions and additionally an 'oracle ray' launch strategy to depict a lower-bound on the time-accuracy values.

**Results.** We present the time-accuracy in Figure A3 and observe: (i) WiNeRT (orange markers) is significantly faster than the simulators (blue line), demonstrating speed-ups of 11-22× over PyLayers (Amiot et al., 2013) and 6-22× over Wireless Insite (Remcom, 2022). Although the simulators are approximately an upper-bound on the accuracy, we find that WiNeRT can make reasonable trade-offs on accuracy to boost simulation times in certain scenarios; (ii) The baselines we propose in this paper (MLP and kNN) are even faster. MLP (green marker) is the fastest with speed-ups of 538-687×, which can be largely attributed to a simple architecture (3-layer ReLU MLP with 128 hidden units). kNN (red marker) is the second fastest with 79-97× speed-ups over the simulators.

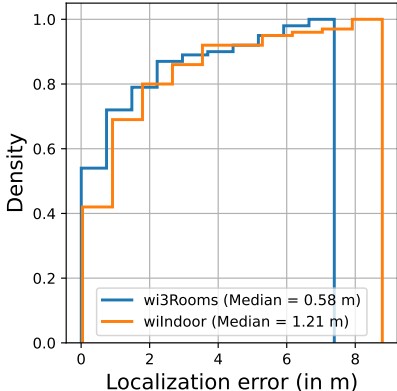

**Figure A4: User Localization.** We backpropagate through our trained forward model to solve for the position of the receiver.

While these baselines offer much faster simulation times, their generalization capabilities remain unclear as they suffer from memorization (see discussion for Fig. 3).

### C.4  USER LOCALIZATION VIA INVERSE RENDERING

In this section, we provide additional details to complement the discussion on the user localization experiment in Section 4.3. For the user localization task, the problem is to determine user location $x_{rx}$ from an observed channel $h_{obs}$. We solve for $x_{rx}$, by performing gradient on spatial coordinate $x_{rx}^{ukn}$ that minimizes the channel loss $\text{render}_\theta(x_{tx}, x_{rx}^{ukn}, F_i)$. This is possible with WiNeRT, since we can backpropagate through the neural simulation of the channel. We optimize for $x_{rx}$ using SGD with momentum (lr=0.01, momentum=0.9, 2000 iterations) with two additional considerations: (a) we constrain $x_{rx}$ to lie in valid ranges (positive, upper-bounded by $x_{max}$) by clamping the values at each iteration; and (b) to prevent solutions in local minimas, we take the result which yields the minimum loss over five random initializations of $x_{rx}$. We present the CDF of localization errors over 100 test examples in A4

## D  IMPLEMENTATION: ADDITIONAL DETAILS

In this section, we provide additional implementation details and hyperparameter choices of approaches discussed in the paper.

### D.1  WINERT

**Architecture: Ray-surface Interaction $f_\theta^1$.** We follow an MLP architecture (see Figure A1) similar to NeRF approaches (Mildenhall et al., 2020; Verbin et al., 2022). We decompose the parameters into view-independent ('spatial MLP') and view-dependent ('directional MLP') sets. Given a ray incident at a spatial co-ordinate $x_k$ in direction $d_k$, the spatial MLP (2 hidden layers, 64 units) takes three inputs: (a) the face $f_i$ (1-hot index) on which $x_k$ lies; (b) the surface normal $n_i$ of face $f_i$; and (c) a 3d vector of signed-distance values between the face and $x_{tx}$, $x_{rx}$, and $x_k$. We find (c) provides information (e.g., $x_{tx}$ and $x_{rx}$ on the same side of wall) to condition the network to predict attributes related to either reflection or transmission components. The directional MLP (1 hidden layer, 64 units) takes two inputs: (i) a 32-dim bottleneck vector produced by the spatial MLP; and (ii) a 3-dim unit vector representing the incidence direction $d_k$. The final output are scaling and additive co-efficients $s$ for the gain magnitude (i.e., $a_k^{(r+1)} = s_1 a_k^{(r)} + s_2$) and 4-dim parameters $\rho_i$ for rotation (based on Euler-Rodrigues formulation). The rotation parameters $\rho_i$ are mapped to a 3×3 rotation matrix $A = \Gamma(\rho_i)$ to transform the incident to outgoing ray $d_k := A d_k$.

**Renderer: Ray Launching.** In the first step of the renderer, we launch $K$ rays from co-ordinate $x_{tx}$ uniformly in all directions. To achieve this, we center a ico-sphere with 5 sub-divisions and

choose as directions the vectors from $\boldsymbol{x}_{\text{tx}}$ towards the ico-sphere vertices (10.2K vertices with 5 sub-divisions). Since we know the exact co-ordinates between $\boldsymbol{x}_{\text{tx}}$ and $\boldsymbol{x}_{\text{rx}}$, we manually include the line-of-sight direction resulting in a total of $K$ rays.

**Renderer: Ray Marching.** The core step of the renderer is ray marching (detailed in Figure 2). We elaborate on technical implementation details step-by-step using as reference Figure 2. We drop sub- and super-scripts for rest of the paragraph for notational convenience. **(a) Ray-Triangle intersection**: For a given ray $\boldsymbol{p} = \boldsymbol{o} + t\boldsymbol{d}$, we are interested in the minimum finite solution of $t > 0$ for which the ray intersects with each face of the mesh. For some face with coordinates $(\boldsymbol{a}, \boldsymbol{b}, \boldsymbol{c})$, this entails solving for $t$ such that $\boldsymbol{p} = \boldsymbol{o} + t\boldsymbol{d} = \alpha\boldsymbol{a} + \beta\boldsymbol{b} + \gamma\boldsymbol{c}$ (under constraints $\alpha + \beta + \gamma = 1$ and $0 \leq \alpha, \beta, \gamma \leq 1$). We calculate valid solutions using Cramer's rule for all faces in the mesh and only consider (if one exists) the minimum positive solution corresponding to the first ray-triangle intersecting point. **(b) Ray-Surface interaction**: Given the solution from the previous step (i.e., on which spatial co-ordinate the ray is incident on the surface), we are now interested in estimating the outgoing ray from that co-ordinate. For this, we leverage an MLP that maps incident gain, direction, and certain face properties to outgoing gain and direction. More details of this MLP are discussed above under the 'Architecture: Ray-surface Interaction'. **(c) Reception/Termination**: Per ray, we stop ray marching steps if it is either received (hits a reception sphere of fixed size of 30cm) or leaves the region of interest (e.g., penetrates exterior wall is shot into infinity). In other cases, we continue with ray marching steps.

**Renderer: Ray Aggregation.** At the end of ray marching steps (over $R$ iterations), we determine the final state of the $K$ rays. We are now interested in a small subset of these $K$ rays that is received at a receiver at fixed co-ordinate $\boldsymbol{x}_{\text{rx}}$. Note that we perform these steps only at test-time. The ray aggregation as a result involves two steps: **(a) Ray Filtering**: where we determine the subset of rays that arrives at $\boldsymbol{x}_{\text{rx}}$ by modelling the receiver as a sphere of fixed radius of 30cm; and **(b) Preventing double counting**: we find duplicate rays arrive at $\boldsymbol{x}_{\text{rx}}$ due to a combination of a non-infinitesimally sized reception sphere and a high density of launched rays. We cull such duplicates by grouping rays based on a unique interaction sequence (i.e., IDs of faces it intersects with) and choosing the ray of the shortest length in each group.

**Optimization.** We perform gradient-descent steps on learnable parameters using Adam with a learning rate of 0.001 with batch size of 1. We observed large gradients (possibly due to single-batch) and hence clip gradient values to 100 during training. The model is trained for 100 epochs and we pick the checkpoint with lowest validation error during training.

## D.2 BASELINES

**MLP.** The MLP baseline extends ideas presented in Tancik et al. (2020); Sitzmann et al. (2020), where a simple MLP is used to map co-ordinates to the signal (e.g., pixel co-ordinate to RGB values). In our paper, the MLP directly maps the spatial co-ordinates $\boldsymbol{x}_{\text{tx}}$ and $\boldsymbol{x}_{\text{rx}}$ to channel $\boldsymbol{h}_i$. The MLP contains 3 hidden layers, each with 128 hidden units and ReLU activation. The core idea here is to implicitly learn the geometry of the environment (floormap $\boldsymbol{F}$), which is common to all train and test examples. Note that in contrast to previous works, this model does not use positional embeddings nor sinusoidal activations, as our initial experiments indicated they learn high-frequency artifacts that is not typically present in our datasets (the wireless channels).

**kNN.** The kNN baseline (with $k$=1) works as so: for a given test-example $(\boldsymbol{x}_{\text{tx}}, \boldsymbol{x}_{\text{rx}})$ we find the spatially closest training example $\arg\min_i ||\boldsymbol{x}_{\text{tx}} - \boldsymbol{x}_{\text{tx},i}^{\text{train}}||_2 + ||\boldsymbol{x}_{\text{rx}} + \boldsymbol{x}_{\text{rx},i}^{\text{train}}||_2$ and predict channel $\boldsymbol{h}_i$.

