# OpenReview forum: "WiNeRT: Towards Neural Ray Tracing for Wireless Channel Modelling and Differentiable Simulations"
_ICLR.cc/2023/Conference — ICLR 2023 poster_

### Official Review · Reviewer_emnY · 2022-10-19

**Confidence:** 2
**Correctness:** 3
**Technical Novelty And Significance:** 3
**Empirical Novelty And Significance:** 3
**Recommendation:** 6

**Clarity, Quality, Novelty And Reproducibility:**

This paper is well-written, and the overall quality is good. I think it is novel and reproducible.

**Strength And Weaknesses:**

Strength:
1. This paper is well-written and the ideas are clearly illustrated.
2. It is interesting to find the similarity between wireless channel modeling and neural ray tracing.
3. It is good to see that WiNeRT achieves encouraging results on time-of-flight prediction and user location tasks.

Weakness:
1. For the baseline method, the paper only compares with KNN and MLP. What are the comparative results with other channel modeling methods?
2. What is the efficiency of the proposed method?
3. Are WI3ROOMS and WIRPLAN two standard benchmarks?
4. In figure 4, the legends and the rays have different colors, it is difficult to know which are GT and which are predictions.


**Summary Of The Paper:**

This paper proposes to employ a differentiable neural ray tracer for wireless channel modeling. It models the time-angle channel impulse response as a superposition of multiple paths, and the wireless characteristics of each path are a result of multiple evaluations of an implicit neural network. The proposed framework achieves strong performance in multiple tasks.

**Summary Of The Review:**

I am not an expert in wireless channel modelling, therefore it is not straightforward for me to evaluate the significance of this work in this field. Given the encouraging results and novelty, I think this paper is above the borderline.

---

> ### Author Response · Authors · 2022-11-18
> **Author Response to Reviewer emnY**
>
> We thank the reviewer emnY for the constructive feedback despite the paper being out of domain expertise. Overall, we are glad that the reviewers generally find the problem/method novel (sfBb, oi9G, YoPQ, emnY), the paper clearly written (sfBb, oi9G, emnY), and accompanied with good evaluation results (sfBb, oi9G, emnY). We equally appreciate the concerns and suggestions raised by reviewers, which we try to address here and by revising the paper (revised text appear in blue). Now, we address the reviewer emnY’s concern and also indicate if it shared by fellow reviewers.
>
> **(common concern - YoPq, emnY) “paper compares with kNN and MLP … no results with other SOTA methods?”**
> No baseline methods exist in literature as we propose the novel problem (sfBb, YoPQ) of modelling neural surrogates for wireless ray tracing. As a result, we develop competitive baselines inspired by surrounding literature: (a) *MLP* is in the spirit of co-ordinate MLPs (Tancik et al. 2020, Sitzmann et al. 2020) which map co-ordinates to signal values; and (b) *kNN* as they have shown to be a strong approach in wireless problems (Sobehy et al., 2020).
>
> **(common concern – sfBb, oi9G, emnY) “What is the efficiency of the proposed method?”**
> We thank the reviewers for their constructive suggestion. In short, we find our approach is efficient both in terms of computation time (6-22x faster than existing simulators) and storage (<324KB to store trained model). We summarize the simulation time experiment in the main paper (Sec 4.4) and further detail in appendix (Sec. C3). However, note that simulators are typically proprietary executables and it is challenging to benchmark runtime complexities on equal grounds.
>
> **“Are WI3ROOMS and WIRPLAN two standard benchmarks?”**
> No, since the problem has not been tackled before.
>
> **“Fig. 4 … the legends and the rays have different colors”**
> Thanks for pointing it out. We fixed it.

---

> ### Author Response · Authors · 2022-12-12
> **Addressing concerns**
>
> Dear Reviewer emnY,
>
> Thank you again for your comments. As the discussion period is closing, we are looking forward to your comments on our rebuttal. We appreciate your valuable time and efforts in reviewing our work.
>
> Some key take-aways from the review and rebuttal:
>   * **Results with other competing approaches**: We make the first step towards a wireless neural simulator and no competing approaches exist to make a fair comparison.
>   * **What is the efficiency?**: We found our approach efficient both in terms of simulation-speed (6-22x faster) and storage (320KB model parameters). More details on Sec. 4.4 and C3.
>
> Best regards,
> Authors

---

### Official Review · Reviewer_YoPQ · 2022-10-21

**Confidence:** 2
**Correctness:** 3
**Technical Novelty And Significance:** 3
**Empirical Novelty And Significance:** Not applicable
**Recommendation:** 6

**Clarity, Quality, Novelty And Reproducibility:**

I think the paper writing is generally clear, but can be improved with more ablation studies about the proposed architecture of neural network. The experimental section should also include some real world scenarios. As pointed out above, how sensitive/robust is the proposed method towards the scene reconstruction noise/errors?

**Strength And Weaknesses:**

Strength: The application of using NeRF-based method for wireless channel modelling is somewhat novel and the proposed implicit method for ray surface interaction seems to be effective.

Weakness: The proposed method seems to assume the reconstructed scene is perfect without any noise. However, in practice, such ideal reconstruction is rarely happens especially for sparse input setting. The proposed method is evaluated using only synthetic data which also raise a concern about the robustness of the proposed method when it is applied to real world scenarios. How does the proposed method avoid overfitting when the training and testing are both under ideal configurations? The proposed method is only compared with kNN and MLP which are very simple baseline methods. How about the comparisons with other SOTA that using explicit methods for ray surface interaction modelling?

**Summary Of The Paper:**

This paper proposed a method for wireless channel modelling inspired from NeRF-based neural scene representation. Different from previous works which explicitly model the ray surface interaction, it tries to learn and map the environment configuration to a wireless channel directly through a neural network. The proposed method is evaluated using multiple indoor scenes and demonstrate significant improvements over the baselines.

**Summary Of The Review:**

Please check my comments above. I think the studied problem and the proposed method is somewhat novel, but the experimental evaluations are lacking in some aspects such as lack of ablation studies on network architectures, lack of robustness of the proposed method towards scene noise, lack of real world scenario examples.

---

> ### Author Response · Authors · 2022-11-18
> **Author Response to Reviewer YoPQ**
>
> We thank reviewer YoPQ for providing an insightful review of our paper. Overall, we are glad that the reviewers generally find the problem/method novel (sfBb, oi9G, YoPQ, emnY), the paper clearly written (sfBb, oi9G, emnY), and accompanied with good evaluation results (sfBb, oi9G, emnY). We equally appreciate the concerns and suggestions raised by reviewers, which we try to address here and by revising the paper (revised text appear in blue). Now, we address reviewer YoPQ’s concern and also indicate if it shared by fellow reviewers.
>
> ### Simulated vs. Real data
>
> **“assumes reconstructed scene is perfect … ideal reconstructions rarely happens”**
> The reviewer is right in pointing out that reconstructed scenes are often noisy. However, this is typically not a major bottleneck for wireless ray tracing where receive power is dominated by rays interacting with large surfaces (e.g., reflections from walls) which are easier to reconstruct. Nonetheless, we believe our approach provides an initial step to work towards probabilistic surrogates that can better handle uncertain reconstructions.
>
> **“evaluated only synthetic data … concern about robustness in real-world scenarios … avoid overfitting when train/test are ideal”**
> It would be ideal to evaluate the model on *real measurement data*. However, solving the problem first on *simulator data* has its own merits: (a) neural surrogates alleviate certain drawbacks of ray tracers (more details in Sec. 1, 3rd paragraph) such as complexity and differentiability; and (b) as we can control simulation parameters (e.g., materials), certain fundamental phenomenon of the real-world (e.g., material-dependent effects) can be isolated for modelling and analysis.
>
> ### Experimental section
>
> **(common concern - YoPq, emnY) “paper compares with kNN and MLP … no results with other SOTA methods?”**
> No baseline methods exist in literature as we propose the novel problem (sfBb, YoPQ) of modelling neural surrogates for wireless ray tracing. As a result, we develop competitive baselines inspired by surrounding literature: (a) *MLP* is in the spirit of co-ordinate MLPs (Tancik et al. 2020, Sitzmann et al. 2020) which map co-ordinates to signal values; and (b) *kNN* as they have shown to be a strong baseline in wireless problems (Sobehy et al., 2020).
>
> **“experimental section lacking … lack of ablation on network architectures”**
> It is unclear how to meaningfully ablate our WiNeRT network architecture, which is a 3-layer MLP integrated in a rendering algorithm. For general ablation, we discuss a baseline “MLP” (without explicit rendering) in Sec. 4.3 and find that WiNeRT outperforms the baseline.

---

> ### Author Response · Authors · 2022-12-12
> **Addressing concerns**
>
> Dear Reviewer YoPQ,
>
> Thank you again for your comments. As the discussion period is closing, we are looking forward to your comments on our rebuttal. We appreciate your valuable time and efforts in reviewing our work.
>
> Some key take-aways from the review and rebuttal:
>   * **Simulated vs. real data**: While evaluating on real data has its own merits, we take a first step towards a neural wireless ray tracing model on simulated data which has its own merits e.g., tackles certain drawbacks (complexity, non-differentiability) of professional ray tracers, allows us to control parameters (materials, geometry) so that we can better analyze and interpret findings.
>   * **Lacking baselines and ablations**: No baseline approach exists to make a fair comparison. Our baselines are designed to ablate the proposed approach e.g., “MLP” baseline implicitly renders the scene.
>
> Best regards,
> Authors

---

### Official Review · Reviewer_oi9G · 2022-10-22

**Confidence:** 3
**Correctness:** 3
**Technical Novelty And Significance:** 3
**Empirical Novelty And Significance:** 3
**Recommendation:** 6

**Clarity, Quality, Novelty And Reproducibility:**

The clarity of the writing and the quality and the novelty of the paper are good. There are some issues with reproducibility since the method contains both learned and non-learned parts and not all details are sufficiently described in the paper.


**Strength And Weaknesses:**

Strengths:

—-------------------

The paper has many strengths and is tackling an important problem. For the sake of time  I am only writing about weaknesses in this review, since those are the ones that should be actioned upon.

Weaknesses (and questions) (W)

—--------------------------------------------

W: The method is a hybrid method that utilizes both typical ray tracing and neural ray marching features. It would be good if the authors could write more precisely which parts of the algorithm are same/similar as normal full ray-tracing and which parts contain learned functions. As far as I understand the main learned part is the Ray-Surface Interaction, and it would be important to describe how this is normally done to understand what the neural surrogate is approximating.

W: Give more details on the end-to-end model, e.g., in an appendix for reproducibility. As it stands it might be difficult to re-implement or improve the current method.

W: The authors should describe more clearly what are the target use cases for the system. It seems to be that the system requires training data from the same/similar environment from an existing ray tracer, so what would be the biggest reasons for using the neural surrogate since an existing ray tracing simulation is anyway needed?

W: The main reason (in addition to inverting) of using a neural surrogate is to improve the inference time simulation speed and computational complexity. As far as I can see this discussion is missing from the paper and it would be very important to discuss it.

W: (See above). If the computational complexity is not reduced much, then the main benefit of the model is the invertibility, which is briefly discussed in the localization experiment. How good is the localization result? It should be compared to other localization approaches in the same environment (without using the trained model). In addition, it would be important to discuss more use cases of the model inversion and more inversion experiments of the neural surrogate should be performed.

W: The paper might find a relatively narrow audience in machine learning and might be even more suitable for publishing in the wireless domain. For ICLR, the authors should discuss which parts could be generalizable to other domains than wireless ray tracing.

W: While there are extensive tests of generalization in the same space as in training (different Rx and Tx locations, etc.), the generalization to new spaces would be interesting to study further in addition to F′ in Figure A3. It will be important to see also the failure cases where there is a big enough change in the environment in the test set, ie., how general ray tracing algorithm it has learned.

W: The authors do not discuss non-linear effects of the channel, such as doppler shift or spread and how this could be modeled for highly mobile UEs.

W: The baselines 1-NN and MLP should be described more carefully. Are they replacing the ray-marching part or the general processing. What parts of a normal ray-tracer are the baselines utilizing (different from the main model)

W: How is diffraction handled in the model?

W: How is penetration handled in the model?

W: Regarding inverse problems, could the authors state which kinds of inverse problems would be solvable using the model and which would lie outside of the current model. For example, could one design changes in the environment geometry using the current model or is it currently restricted only to the trained geometry.


**Summary Of The Paper:**

Update; I have read the rebuttal and decided to increase the score.
--------

This paper proposes a neural surrogate (WiNeRT) for a ray tracer that models the propagation of wireless signals. WiNeRT is a hybrid heuristic/learned ray tracer that replaces some heuristic ray tracing functionality, especially the Ray-Surface Interaction with a neural network. The authors claim that this is the first wireless neural ray tracer and in addition to simulating ray-tracing results, it will allow solving inverse problems via differentiating through the model. The paper experimentally validates the accuracy of the model against simple baselines and shows initial results on inverse problem of localization.

**Summary Of The Review:**

Well-written practical paper that has good empirical evaluation, but would need some more work. Once improved and with detailed end-to-end description of the method it might serve also as a baseline for further works.

---

> ### Author Response · Authors · 2022-11-18
> **Author Response to Reviewer oi9G (1/2)**
>
> We thank reviewer oi9G for providing a thorough review of our paper. Overall, we are glad that the reviewers generally find the problem/method novel (sfBb, oi9G, YoPQ, emnY), the paper clearly written (sfBb, oi9G, emnY), and accompanied with good evaluation results (sfBb, oi9G, emnY). We equally appreciate the concerns and suggestions raised by reviewers, which we try to address here and by revising the paper (revised text appear in blue). Now, we address reviewer oi9G’s concern and also indicate if it shared by fellow reviewers.
>
> ### Questions on Target use-case
>
> **“Target use-case to use neural surrogate over ray tracing simulators? Simulation speed? Invertibility? …”**
> There are many use-cases (also mentioned in Sec. 1, 3rd paragraph):
> 1. *Closed-loop design*: to enable end-to-end optimization of metrics (e.g., power, latency) w.r.t design parameters (e.g., base station location and orientation) by “whitening” the intermediary black-box simulator.
> 2. *Complexity reduction*: our approach results in significant speed-ups (6-22x faster; Sec. 4.4) and has a low memory footprint (<324KB to store trained model). This combination shows promise in running simulations on-device.
> 3. *Dynamic scenes*: because we model spatially local interactions (Sec. 4.4) and (b), the neural surrogate can be used to simulate dynamic scenes (e.g., moving vehicles) which is cumbersome for classical wireless simulators
> 4. *Inverse problems*: such as for localization (0.58-1.2m localization error; Sec 4.4).
>
> Key to all such use-cases is a surrogate model that reasonably mimics the ray tracer simulations, which we believe we achieve (“good empirical evaluation”-oi9Gm, “solid experiments”-sfBb, “encouraging results”-emnY).
>
> **(common concern – sfBb, oi9G, emnY) “comparison of simulation speeds … discussion missing”**
> We thank the reviewers for their constructive suggestion. In short, we find significant speed-ups (6-22x faster than existing simulators) with our vanilla implementation. We summarize the simulation time experiment in the main paper (Sec 4.4) and further detail in appendix (Sec. C3). However, note that simulators are typically proprietary executables and it is challenging to benchmark runtime complexities on equal grounds.
>
> **“generalization concerns … how general ray tracing is learnt … failure cases with big enough changes in environments”**
> The reviewer raises a valid point on generalization of surrogates on drastic scenario changes (e.g., indoor to outdoor). However, generalization is typically achieved by classical simulators at a cost (e.g., runtime complexity). As a result, our work is particularly motivated by ‘distilling’ scenario-specific simulation knowledge into a neural surrogate model, which has its own merits (e.g., complexity reduction, deployable and learnable on base station). We revised the paper’s introduction to clarify this.
>
> **“invertibility … how good is localization result? Should be compared to other localization results”**
> RF-based user localization has a rich literature and comparing them in our localization use-case experiment would be ideal. However, a direct comparison is out of scope since  localization literature is typically application-dependent (e.g., specific to 5G, wifi) whereas our approach is not. As a result, rather than target localization benchmarks, this experiment is intended to demonstrate that we can backpropagate through the surrogate simulator. We revised the paragraph to prevent confusion.
>
> ### Clarifications
>
> **“how is penetration handled in the model?”**
> The ray-surface interaction captures both penetration and reflection effects. We discuss in Sec 4.4 “What does the ray-surface interaction learn”. Additionally, the effects can be qualitatively observed in supplementary videos.
>
> **“ how is diffraction handled in the model?”**
> We do not model diffraction. This is an interesting open-problem and we mention it under “Limitations and Future Work” (Sec. 5).

---

> > ### Comment · Reviewer_oi9G · 2022-12-12
> > **Thank you for the rebuttal.**
> >
> > I have read the rebuttal and since many of my concerns are addressed I have decided to increase the score.

---

> ### Author Response · Authors · 2022-11-18
> **Author Response to Reviewer oi9G (2/2)**
>
> ### Writing additional details in the paper
>
> **“not all details are sufficiently described in the paper ... some issues with reproducibility”**
> Thanks for pointing this out. Although reviewers find our work reproducible (sfBb, emnY), we made more passes over the paper and additionally aggregated relevant details in Section D.
>
> **“more details on … parts of algorithms that is learnt vs. similar to ray-tracing”**
> In addition to our previous color-coding of parts as “learnable” and “non-learnable” in Fig. 2, we further elaborate this distinction in the figure caption.
>
> **“more details on … how is ray-surface interaction normally done”**
> Thanks for the suggestion. Ray-surface interaction effects are normally calculated by looking-up material values from standard databases. We added details in Sec. 3.2.
>
> **“more details on … end-to-end model, 1-NN and MLP”**
> Great suggestion. We aggregate implementation details from the main paper and further elaborate them in Sec. D1 (WiNeRT) and Sec. D2 (1-NN, MLP).
>
> **“relatively narrow audience in ML … suitable for wireless domain … discuss which parts could be generalizable to other domains”**
> We believe our approach is suitable for other domains (“… bring more impact to computer vision…”-sfBb), as it builds on NeRF-based principles. Furthermore, many physics-based neural simulations are typically published in ML venues (more details in Sec. 2). Nonetheless, we understand the reviewer’s concern and include a “Broader Technical Impact” paragraph in Sec. 5 and discuss interplay with non-wireless modalities (e.g., spatial acoustics).
>
> **(common concern – sfBb, oi9G) “discuss non-linear effects of the channel”**
> This is an interesting idea. While wireless propagation is pre-dominantly modeled as a linear system (Tse and Viswanath, 2005), such models (including our own) fail to capture non-linear effects (e.g., amplifier saturation). This is a limitation of our current work and we now discuss under “Limitations and Future Work” (Sec. 5).

---

> ### Author Response · Authors · 2022-12-12
> **Addressing concerns**
>
> Dear Reviewer oi9G,
>
> Thank you again for your comments. As the discussion period is closing, we are looking forward to your comments on our rebuttal. We appreciate your valuable time and efforts in reviewing our work.
>
> Some key take-aways from the review and rebuttal:
>   * **Target use-case**: There are multiple use-cases (e.g., exploiting differentiability, complexity reduction) enabled by a differentiable wireless renderer. The focus of the paper is the first-step towards it: how to neurally model such a render to empricially mimic the performance of a commercial ray tracer? The paper further presents proof-of-concept of the use-cases e.g., complexity reduction (6-22x faster with a 320KB model), inverse problems (user localization with 0.58m-1.21m errors).
>   * **Additional details in paper**: We added many additional details based on the suggestions (e.g., more details on NN vs non-NN parts, baselines)  – thanks once again.
>
> Best regards,
> Authors

---

### Official Review · Reviewer_sfBb · 2022-10-24

**Confidence:** 2
**Correctness:** 4
**Technical Novelty And Significance:** 4
**Empirical Novelty And Significance:** 4
**Recommendation:** 8

**Clarity, Quality, Novelty And Reproducibility:**

This paper has clearly described the motivation of wireless signal propagation, the necessity of simulating wireless signals. It also has provided a comprehensive literature review, although no previous research in this line.  The method is clearly written with the aid of using mathematical formulas. The novelty is quite strong because this is a new topic with no pre-existing solutions.

The code is provided and the reproducibility and originality of the work is good.

**Strength And Weaknesses:**

Strengths
1. This paper creates a new area that is simulating / predicting wireless signal propagation problem (or more generally: a ray-tracing simulation). This method explores a new topic and expands the boundary of computer vision applications.
2. The proposed paper has thorough study with signal propagation problem. It provides detailed mathematical definition of the problem, casts the physics ray-tracing computation to a network simulation problem. It also come with solidate experiments to demonstrate the effectiveness of the proposed method.
3. The paper also proposes two new databases / datasets that allow the community to benchmark further new methods. This could bring more impact to the community of computer vision.

Weaknesses
1. This paper does not discuss about the non-linear surface or interactions of wireless signals. It assumes the operations are all linear. The authors are suggested to provide a limitation of the method
2. In the introduction part, the goal of the proposed method is to reduced the inference time. The authors are suggested to make a comparison between the current method and ray-tracing computation.

**Summary Of The Paper:**

This paper proposes a neural network based solution to heuristically solve a wireless signal (physics) rendering problem. Given the environment set-up and configurations of the transmitter and receivers, the pre-trained network is able to simulate the wireless signal propagation in the confined environment instead of physically computing the ray-tracing function, which costs the computational power extremely. In addition, this approach enables the reverse rendering applications.



**Summary Of The Review:**

In summary, this paper proposes a new topic of simulating wireless signal propagation in a confined configurable environment. The paper systematically defines and solves the problem with comprehensive experiments.

---

> ### Author Response · Authors · 2022-11-18
> **Author Response to Reviewer sfBb**
>
> We appreciate the highly constructive and balanced review of reviewer sfBb. Overall, we are glad that the reviewers generally find the problem/method novel (sfBb, oi9G, YoPQ, emnY), the paper clearly written (sfBb, oi9G, emnY), and accompanied with good evaluation results (sfBb, oi9G, emnY). We equally appreciate the concerns and suggestions raised by reviewers, which we try to address here and by revising the paper (revised text appear in blue). Now, we address reviewer sfBb’s concern and also indicate if it shared by fellow reviewers.
>
> **(common concern – sfBb, oi9G) non-linear interactions not captured … provide a limitation**
> This is a valid remark. While wireless propagation is pre-dominantly modeled as a linear system (Tse and Viswanath, 2005), such models (including our own) fail to capture non-linear effects (e.g., amplifier saturation). This is a limitation of our current work and we now discuss under “Limitations and Future Work” (Sec. 5).
>
> **(common concern – sfBb, oi9G, emnY) comparison of simulation speeds … discussion missing**
> We thank the reviewers for their constructive suggestion. In short, we find significant speed-ups (6-22x faster than existing simulators) with our vanilla implementation. We summarize the simulation time experiment in the main paper (Sec 4.4) and further detail in appendix (Sec. C3). However, note that simulators are typically proprietary executables and it is challenging to benchmark runtime complexities on equal grounds.

---

> ### Author Response · Authors · 2022-12-12
> **Addressing concerns**
>
> Dear Reviewer sfBb,
>
> Thank you again for your comments. As the discussion period is closing, we are looking forward to your comments on our rebuttal. We appreciate your valuable time and efforts in reviewing our work.
>
> Some key take-aways from the review and rebuttal:
>   * **Non-linearity**: We model the wireless channel as a linear system much like existing literature. Nonetheless, this is a limitation and we remark in Sec 5.
>   * **Simulation speeds**: We find 6-22x speed-ups and further discuss in Sec. 4.4 and Sec. C3.
>
> Best regards,
> Authors

---

### Decision · Program_Chairs · 2023-01-20

**Decision:**

Accept: poster

**Justification For Why Not Higher Score:**

Honestly, it would be unfair to the authors of the paper to state an arbitrary reason for not attributing a higher score. The paper has a significant potential to be impactful as it introduces the first differentiable neural ray tracer for wireless channel modelling and new datasets for benchmarking purposes are provided. As a meta-reviewer who tries to be fair, objective and intellectually honest, the only reason for which the paper has not been considered for the spotlight or oral is that it has a grade of less than 7.

Determining if a paper should receive an award or a mention in a venue according to some arbitrary criterion could be meaningless. In the history of machine learning, great impactful foundational works, such as Generative Adversarial Networks (Goodfellow et al, 2014), were not accepted as spotlights or orals and yet their long-term impact was the true reward that authors could reap.

- References:
Goodfellow, Ian, et al. "Generative adversarial networks." (NeurIPS 2014)

**Justification For Why Not Lower Score:**

Reviewers unanimously recommended acceptance of the paper and most reviewers agreed that the paper is marginally above threshold. The paper has the potential to be impactful as it introduces the first differentiable neural ray tracer for wireless channel modelling and new datasets for benchmarking purposes are provided.

**Metareview: Summary, Strengths And Weaknesses:**

I Summary:

- I.1 Investigated Problem:

 The paper investigates the problem of wireless rendering (e.g., times-of-flight, power of each path) in an environment as a function of the sensor's spatial configuration (e.g., placement of transmit and receive antennas). Unlike classical ray tracers, a surrogate model is proposed to take into account elements of statistical modelling and learns directly on field data. A neural surrogate WineRT is THEN proposed to model wireless electromagnetic propagation effects in indoor environments.

- I.2 Proposed Solution:
WiNeRT is a neural surrogate to classical ray tracers that provides a representation of the environment that is:
	- fast;
	- differentiable: enables end-to-end optimization for downstream tasks (e.g., network planning);
	- continuous;
	- able to task an MLP to learn how surfaces in a 3D environment influence propagation of wireless rays, (e.g., predicting the attenuation factor of a reflective component)

- I.3 Validity Proof of the Proposed Solution:
Empirical evidence is provided to support the validity of the proposed solution:
    - The proposed method is evaluated using multiple indoor scenes and demonstrates significant improvements over the baselines.
     - Results are also provided on time-of-flight prediction and user location tasks.

II. Strengths:

- II.1 From a structural point of view:
    - The paper is well-structured and the ideas are clearly illustrated.

- II.2 From an analytical point of view:
    - The proposed method explores a new topic and expands the boundary of computer vision applications;
    - improves the inference time simulation speed and computational complexity.
    - Authors propose two new datasets to benchmark further new methods. This could bring more impact to the community of computer vision.

- III From a Potential perspective (Potential of the paper to the community):
    - The proposed solution has a great potential to be of benefit to the whole community. The paper introduces the first differentiable neural ray tracer for wireless channel modelling and new datasets for benchmarking purposes are provided. The potential benefit of the work is immediate and concrete.

- IV Weaknesses:
    - Most of the authors pointed out that the paper does not discuss non-linear effects of the channel, such as doppler shift or spread and how this could be modeled for highly mobile UEs. The authors acknowledged during the rebuttal that wireless channel is modeled as a linear system which conforms to the standard practice in the existing literature. Nevertheless, the authors recognized that is a limitation and added a related subsection in the main text.


**Note From Pc:**

if the above contains the word "oral" or "spotlight" please see: "oral" presentation means -> notable-top-5% and "spotlight" means -> notable-top-25%. As stated in our emails, we are disassociating presentation type from AC recommendations

**Summary Of Ac-Reviewer Meeting:**

N/A